# Preclinical Development of a Bacteriophage Cocktail for Treating Multidrug Resistant *Pseudomonas aeruginosa* Infections

**DOI:** 10.3390/microorganisms9092001

**Published:** 2021-09-21

**Authors:** Sophie Camens, Sha Liu, Karen Hon, George Spyro Bouras, Alkis James Psaltis, Peter-John Wormald, Sarah Vreugde

**Affiliations:** 1Department of Health and Medical Sciences-Surgery, The University of Adelaide, Adelaide 5000, Australia; sha.liu@adelaide.edu.au (S.L.); karen.hon@adelaide.edu.au (K.H.); george.bouras@adelaide.edu.au (G.S.B.); alkis.psaltis@adelaide.edu.au (A.J.P.); peterj.wormald@adelaide.edu.au (P.-J.W.); Sarah.vreugde@adelaide.edu.au (S.V.); 2Department of Surgery-Otolaryngology Head and Neck Surgery, Basil Hetzel Institute for Translational Health Research, The Queen Elizabeth Hospital, Woodville South 5011, Australia

**Keywords:** bacterial infection, bacteriophage, *Pseudomonas aeruginosa*, cystic fibrosis, multidrug resistance, phage therapy

## Abstract

A *Pseudomonas aeruginosa* (*P. aeruginosa*) airway infection is one of the predominant causes contributing to the high morbidity and mortality rates in cystic fibrosis (CF) patients. The emergence of antibiotic resistant *P. aeruginosa* strains has led to an urgent need for new therapeutic approaches. Bacteriophages (phages) are viruses that can infect and lyse specific bacteria, providing a potential alternative approach in targeting antibiotic-resistant strains. We aim to isolate and characterise novel *P. aeruginosa* phages for combination in a cocktail to kill *P. aeruginosa.* One particular phage, PA4, could lyse 14/20 clinical isolates as observed through spot assays. This phage could significantly reduce the growth of bacteria in vitro, as determined through planktonic adsorption and inhibition assays as well as crystal violet- and LIVE/DEAD-stained biofilm assays. A morphological and genomic analysis revealed that PA4 belongs to the Myoviridae family and contained 66,450 bp. The broad infectivity profile, good stability in various pH and temperature conditions, lytic ability and the absence of the absences of antibiotic resistance, toxic and lysogenic genes suggest that PA4 is a good candidate for clinical grade use. Overall, phage therapy represents a promising alternative treatment option to antibiotics when treating a *P. aeruginosa* infection.

## 1. Introduction

*Pseudomonas aeruginosa (P. aeruginosa*) is a gram-negative bacterium, considered to be an opportunistic pathogen [1,2]. It is ubiquitous in the environment and is frequently responsible for causing nosocomial infections in humans [1,2]. *P. aeruginosa* represents one of the most common bacterial pathogens in an upper and a lower airway infection and is of particular significance in cystic fibrosis (CF) patients [1]. CF patients lack a functioning CF transmembrane conductance regulator gene, resulting in a change to the airway mucosa that is favourable to *P. aeruginosa* infection [2]. It is one of the first pathogens to infect and dominate the airways during a chronic infection and has been associated with worsening lung function in CF patients [2,3]. CF patients are also frequently affected by chronic rhinosinusitis (CRS), in which colonisation of the paranasal sinuses with *P. aeruginosa* has been found to be a reservoir for a recurrent lung infection [1].

*P. aeruginosa* has the capacity to produce diverse and difficult to treat infections due to the release of virulence factors and the formation of a biofilm [2,4]. Phenotypic changes in the infecting strain over time result in the conversion to antibiotic-resistant phenotypes, which have been correlated with increased morbidity and mortality [2]. The primary treatment modality for CF patients involves systemic and inhaled broad-spectrum antibiotics from early in their life, which can lead to the development of multidrug-resistant (MDR) strains of *P. aeruginosa* [2,3,5]. Due to the emerging threat of infections with antibiotic-resistant strains and the lack of research and development of new antibiotics, there is an urgent need for new treatment options that are effective against such infections [6,7].

One potential treatment option that is being investigated as an alternative or complimentary treatment to antibiotics is bacteriophage (phage) therapy [8,9]. Phages were discovered over a century ago and have recently regained interest for their potential to treat MDR infections [10,11]. Lytic phages are viruses that infect, replicate within and eventually lyse bacteria, causing cell death [1]. *P. aeruginosa* phages have shown promise in several in vitro and in vivo preclinical studies [1,12,13,14]. Some advantages of phage therapy compared to conventional antibiotics include the ability to target specific bacterial species, activity against antibiotic-resistant strains, replication at the site of infection and biofilm eradication [1,2]. Phages are species specific, and therefore target the pathogenic bacteria without disturbing non-harmful commensal bacteria, resulting in less systemic side effects than antibiotics [1,15]. Due to this specificity, treating bacterial infections requires access to phages that are biologically active against the patient’s bacterial strains [4]. CF patients are often infected with several *P. aeruginosa* strains that coexist within a patient, therefore, the development of a phage cocktail will increase the antibiofilm activity by expanding the target host range and decreasing the potential for phage resistance [1,2,12]. *P. aeruginosa* phages have shown to be effective against planktonic and biofilm *P. aeruginosa*, however, the use of phages in Australia is currently not available [1,4,14]. Therapeutic phage candidates will be required to meet quality and safety requirements for human application [16]. Hence, the isolation and characterisation of phages will follow the recommended minimal requirements for sustainable phage therapy products as proposed by Pirnay et al. as a guide towards the clinical application [16]. There is an urgent need for the development of new treatment options that are effective at reducing infections in CF patients, and phage therapy presents a promising option in targeting *P. aeruginosa* infections [1,4].

## 2. Materials and Methods

### 2.1. Bacterial Strains and Growth Conditions

The Central Adelaide Local Health Network Human Research Ethics Committee approved the use of clinical isolates for this study (HREC/15/TQEH/132). *P. aeruginosa* reference strain ATCC15692 (PA01) was obtained from the American Type Culture Collection (Manassas, VA, USA). The *P. aeruginosa* clinical isolates of patients with CF (n = 9) were kindly donated by the Department of Otorhinolaryngology, Academic Medical Centre (Amsterdam, The Netherlands) and the *P. aeruginosa* clinical isolates from non-CF (n = 11) isolated from the sinonasal cavities of CRS patients with and without nasal polyps were kindly donated by the Department of Otolaryngology-Head and Neck Surgery, University of Adelaide at The Queen Elizabeth Hospital (TQEH; Woodville, SA, Australia). The presence of *P. aeruginosa* had been confirmed and the bacteria were isolated by Adelaide Pathology Partners (Mile End, SA, Australia). *P. aeruginosa* clinical isolates had been confirmed again using Cetrimide selective agar plates (Oxoid, Thebarton, SA, Australia) and an oxidase reagent test (BioMérieux, Marcy l’Etoile, France). The clinical isolates used to characterise phages were further confirmed using MALDI-TOF (Bruker, VIC, Australia).

The *P. aeruginosa* clinical isolates (n = 20) were stored in 25% glycerol in tryptone soya broth (TSB) (Oxoid, Thebarton, SA, Australia) at −80 °C. The clinical isolates were plated from frozen glycerol stocks onto 1.5% tryptone soya agar (TSA) plates and broth cultures were grown in TSB (Oxoid, Thebarton, SA, Australia). Unless otherwise stated, the agar plates and broth cultures for non-CF clinical isolates were incubated at 37 °C (Orbital Mixer Incubator, Ratek, Boronia, VIC, Australia) and the CF clinical isolates were incubated at 37 °C supplied with 5% extra CO_2_ (Panasonic Healthcare Co., Tokyo, Japan) with 180 rpm shaking for the broth cultures.

### 2.2. Bacteriophage Isolation

The *P. aeruginosa* phages (n = 15) were obtained from TQEH wastewater. Wastewater samples were collected at 5 different time points every 30–60 min from the palliative care, respiratory, dermatology and infectious disease ward and intensive care unit departments. Samples of 50 mL from each time point were centrifuged at 4000 rpm for 10–30 min to remove cellular debris and faecal matter (Allegra X-30R Centrifuge, Beckman Coulter, NSW, Australia). The supernatant was filtered using a 0.2 μm filter (PALL Acrodisc, NY, USA) to remove the bacteria and debris. An amount of 0.5 McFarland Units (McF) of PA01 diluted 1:100 in TSB was grown overnight at 37 °C with 180 rpm shaking. An amount of 100 μL of filtered solution was combined with 100 μL of the PA01 overnight culture and 4 mL of molten (0.4%) TSA and plated on 1.5% TSA plates via the double agar overlay method as described [17]. Following overnight incubation, the formation of plaques confirmed the presence of phage. Individual plaques with different morphology were picked using pipette tips and placed into a 1.5 mL screw neck vial (Thermo Fisher Scientific, Waltham, MA, USA) containing sodium magnesium buffer (SM; 100 mM sodium chloride, (Oxoid, Thebarton, SA, Australia); 8 mM magnesium sulfate heptahydrate, (Sigma-Aldrich, Castle Hill, NSW, Australia); 50 mM 1 M Tris HCL, pH 7.5), respectively, and vortexed vigorously before storage for 2–6 days at 4 °C for propagation. 

### 2.3. Bacteriophage Propagation

A two-step propagation was applied to amplify and purify the isolated phage. For small-scale amplification of phage, 50 μL of PA01 overnight culture was added into 5 mL TSB and incubated for 1 h. An amount of 100 μL of phage in SM buffer was added into the coculture and incubated overnight. The supernatant containing phage was harvested after centrifugation (4000 rpm, 30 min) and filtration (0.2 μm syringe filter). Phage titration was performed before large-scale amplification.

For large-scale amplification of phage, 500 μL of the PA01 overnight culture was incubated with 50 mL of TSB for 1 h. Phage was then added into the coculture at a multiplicity of infection (MOI) = 1. The phage lysis was harvested as described above. 

### 2.4. Bacteriophage Titration and Concentration

The phage titer was determined via a double agar spot assay as a modified protocol [17]. Briefly, 100 μL of the PA01 overnight broth culture was added to 4 mL of molten (0.7%) TSA and overlaid onto a 1.5% TSA plate. Phages were serially diluted in SM Buffer and spot assays were performed in triplicate with a drop size of 3 μL. The plates were incubated at 37 °C overnight. The SM buffer solution alone was assessed as a negative control. The phage titer was counted and calculated. The samples requiring a higher titer were concentrated using a 100k MWCO Pierce^TM^ Protein Concentrator PES (Thermo Fisher Scientific, Waltham, MA, USA) according to the manual.

### 2.5. Bacteriophage Host Range Analysis

The ability of phage isolates to lyse 20 *P. aeruginosa* clinical isolates (9 CF and 11 non-CF) was tested using the double agar spot assay as described [17]. Briefly, 100 μL of overnight culture for each clinical isolate was added to 4 mL of molten (0.7%) TSA and overlaid onto 1.5% TSA plates. Phage isolates at 1 × 10^9^ plaque forming units per mL (PFU mL^−1^) were serially diluted in SM buffer and 3 μL spots applied in triplicate on top of the plates. The phage sensitivity was determined as described previously [17,18] using the clarity of plaques and classified as sensitive (+) for a clear lysis, semi-sensitive (+/−) for partial lysis and non-sensitive (−) for no lysis. 

### 2.6. Bacteriophage Stability Testing

The stability of isolated phages was tested against a wide pH (3–12) and temperature (−80 °C to +80 °C) range using a working stock in TSB with an initial phage titer of around 1 × 10^9^ PFU mL^−1^. The pH stability testing was studied according to the methods described by Bae et al. [19] Briefly, 10 μL working stock of each phage was suspended in 90 μL SM buffer previously adjusted with 1 M NaOH or HCL (Thermo Fisher Scientific, Winsford, United Kingdom) to yield a pH range of 3–12. The samples were incubated at room temperature (RT) for 1 h. For thermal stability, 80 μL of working stock phage was added to an Eppendorf tube and incubated for 1 h in an Eppendorf Thermomixer Compact (Sigma-Aldrich, Castle Hill, NSW, Australia) at 4 °C, 30 °C, 37 °C, 40 °C, 50 °C, 60 °C, 70 °C and 80 °C. The long-term storage of phage was examined over a 6-month period. An amount of 100 μL of phage at 1 × 10^9^ PFU mL^−1^ previously diluted in TSB were stored at RT, 4 °C, −20 °C and −80 °C. The phage stability for each experiment was determined by measuring the phage titration.

### 2.7. Minimum Inhibitory Concentration (MIC) Assays

The resistance to commonly used antibiotics was determined on aerobic, planktonic cultures using broth microdilution minimum inhibitory concentration (MIC) assays, as described [20]. The antibiotics tested were ciprofloxacin, gentamicin, imipenem, tobramycin, obtained from Sigma-Aldrich (Castle Hill, NSW, Australia) and netilmicin (Chem-Supply, Gillman, SA, Australia). The clinical isolates were designated as being resistant as per the Clinical and Laboratory Standards Institute (CLSI) recommendations (Adelaide Pathology Partners, Mile End, VIC, Australia) [21]. 

### 2.8. Bacteriophage Adsorption Assays

Phages PA4 and PA8 were selected for planktonic testing against 1 CF (C462) and 1 non-CF (C446) clinical isolate along with the PA01 reference strain. The optical density (OD) was measured at 600 nm in a spectrophotometer (SmartSpec 3000, Bio-Rad, CA, USA) for each overnight culture of CF and non-CF *P. aeruginosa* CIs and 400 μL of each overnight culture was diluted 1:100 in fresh TSB in a total volume of 40 mL. Phages were added at MOI = 0.1. The bacterial suspension was incubated at 37 °C with 180 rpm shaking. After taking an initial 3 mL sample at 0 min, 3 mL samples were taken at 1 min, 5 min, 7 min, 10 min and every 10 min thereafter for 40 min. The samples were centrifuged at 4000 rpm for 10 min. The supernatant containing the unadsorbed phages was filtered through a 0.2 μm filter and plated using the double agar spot assay method. The percent of unadsorbed phages of initial phage inoculum at each time interval was calculated. The adsorption experiments were repeated 3 times.

### 2.9. Bacteriophage Inhibition Assays

The infectivity profile of the 2 selected phages was assessed at MOI = 0, 0.1 and 1 using inhibition assays. In brief, the overnight cultures of each clinical isolate were prepared and the required volume of phage stock solution (~10^8^ PFU mL^−1^) was added to the corresponding suspension to achieve an MOI = 0, 0.1 and 1. The samples were incubated at 37 °C with 180 rpm shaking. Every 30 min for 6.5 h, the OD600 value was measured.

### 2.10. Crystal Violet Biofilm Assay

A 1.0 McF suspension in 0.9% saline of the clinical isolate was diluted 1:15 in TSB and gently mixed by inversion. An amount of 150 μL/well of the resulting suspension was plated into a FluoroNunc, U-Shaped 96-well Microplate (Thermo Fisher Scientific, Roskilde, Denmark). Wells adjacent to the top and bottom edge of the plate were filled with 200 μL of sterile phosphate buffer solution (PBS) to prevent dehydration during incubation. Positive and negative controls were added to each plate containing no phage treatment and only TSB solution, respectively. The plates were incubated for 24 h on a gyratory mixer (Ratek, VIC, Australia). Following 24 h, the wells were gently aspirated and washed twice with sterile PBS to remove any remaining planktonic cells. An amount of 180 μL/well of each treatment (phages PA4 and PA8) in TSB each at concentrations of 1 × 10^7^, 10^8^ and 10^9^ PFU mL^−1^ were plated in six replicates and the biofilms were assessed after 24 h incubation. At 24 h post-treatment, the liquid contents of the wells were aspirated and washed gently twice with sterile PBS and stained with 180 μL/well 0.1% crystal violet (Sigma-Aldrich, Castle Hill, NSW, Australia) for 15 min. The stained plates were rinsed with distilled water and left to dry overnight. The crystal violet stain was eluted by the application of 180 μL/well 30% acetic acid (Chem-Supply, Adelaide, SA, Australia) and the plate was incubated at RT for 1 h. The acetic acid was pipetted into a new flat-bottom 96-well plate (BMG Labtech, Ortenberg, Germany) and the absorbance measured at 595 nm using the FLUOstar Optima microplate reader (BMG Labtech, Ortenberg, Germany).

### 2.11. LIVE/DEAD Staining

The biofilm viability was determined using the Invitrogen LIVE/DEAD BacLight Bacterial Viability Kit (Thermo Fisher Scientific, United Kingdom). Briefly, biofilms were grown over 24 h in 8-well cell imaging slides (Eppendorf, Hamburg, Germany) containing no bacteriophage as a positive control. The wells were aspirated and washed gently twice with 0.9% saline before application of 400 μL/well of PA4 treatment in TSB at concentrations of 1 × 10^7^, 10^8^ and 10^9^ PFU mL^−1^. Following 24 h incubation, the wells were aspirated and washed again prior to the application of 180 μL/well 5% glutaraldehyde fixative (Sigma-Aldrich, Castle Hill, NSW, Australia) for 45 min. Prior to staining, the wells were aspirated and washed again. Two stock solutions of stain SYTO9 and propidium iodide (PI) were each diluted in MilliQ water to give a 1 mL:1.5 µL:1.5 µL ratio and 400 µL was applied to each well for 15 min before aspirating and washing the wells. Live SYTO9-stained cells and dead PI-stained cells were visualised with a confocal laser microscope at 20× magnification and the percentage of cell death was determined (Zeiss LSM700, Carl Zeiss AG, Oberkochen, Germany).

### 2.12. Bacteriophage Morphology

The phage morphology for PA4 and PA8 was determined by transmission electron microscopy (TEM) using a modified protocol [22]. Briefly, 5 μL of phage diluted 1:10 in SM buffer was placed on the coated side of a carbon/formvar grid (ProSciTech Pty Ltd., Kirwan, QLD, Australia) for 3 min before being wicked dry with filter paper. An amount of 5 μL of electron microscopy (EM) fixative (1.25% glutaraldehyde, 4% paraformaldehyde in PBS to which 4% sucrose had been added) was then placed on the grid for 2 min before being wicked dry, followed by 5 μL of 2% uranyl acetate for 2 min. The samples were examined in a FEI Tecnai G2 Spirit 120 kV TEM (FEI Technologies Inc., Hillsboro, OR, USA).

### 2.13. Bacteriophage Genomic Sequencing

Genomic sequencing was carried out for PA4 and PA8. A DNA extraction from a 1 mL aliquot of phage (>1 × 10^10^ PFU mL**^−^**^1^) was performed using the Phage DNA Isolation Kit (Norgen Biotek Corp, Thorold, ON, Canada), following the protocol provided by the manufacturer. A genomic sequencing of the isolated DNA was conducted by an external laboratory (SA Pathology, Adelaide, SA, Australia). Paired-end sequence reads were obtained on an Illumina sequencer. Poor reads were trimmed using Trimmomatic [23]. The genome assembly was performed using the De Bruijn graph-based short read de novo assembler ABYSS 2.0 [24]. The sequences were verified using QUAST [25] and annotated with PROKKA [26] and RASTtk [27]. A genomic analysis was conducted with the non-redundant (nr) NCBI database using MEGABLAST. Genome alignments were obtained using Mauve and plotted in R using genoplotR. An individual analysis of the phages included the phage in question and 14–18 closely related *P. aeruginosa* phage genomes available in the nr NCBI database. 

### 2.14. Statistical Analysis

A Statistical analysis was performed using GraphPad Prism v.9 (GraphPad Prism, version 9.0.0 (86); Macintosh Version by Software MacKiev © 2021-2020 GraphPad Software, LLC.; San Diego, CA, USA). To determine the effect of pH, high temperature and long-term storage on phage viability, as well as planktonic and biofilm efficacy, data was analysed using a one-way analysis of variance (ANOVA) followed by Tukey’s and Dunnett’s post hoc multiple comparison test. Significant differences were determined at *p* < 0.05. All of the experiments were repeated three times and performed in triplicate or six replicates. The mean values of the three or six replicates obtained with standard error of the means (SEM).

## 3. Results

### 3.1. Isolation and Host Range Determination of Bacteriophages

From the various wastewater samples collected, fifteen *P. aeruginosa* phages were isolated according to their morphology displayed on the agar plates and designated PA1 to PA15. The initial phage titer was determined after the propagation (Appendix A) for further host range testing. 

The host range against 20 *P. aeruginosa* clinical isolates (9 CF and 11 non-CF) for individual phages varied between 6/20 and 12/20 sensitive (+) strains and 4/20 and 11/20 semi-sensitive (+/−) strains, indicating moderate lytic activity and a broad host range (Figure 1, Table 1). No phages were able to lyse the clinical isolates C453 and C452, therefore displaying no sensitivity (−) (Table 1).

### 3.2. Multidrug Resistance in P. aeruginosa Clinical Isolates

The number of clinical isolates displaying resistance to the five antibiotics tested is shown in Appendix A. Various clinical isolates displayed resistance to some of the three classes of antibiotics commonly used to treat *P. aeruginosa* [28].

### 3.3. Bacteriophage Stability at Acidic and Alkaline pH

All of the phage isolates were tested against a wide pH range (pH 3–12), over a 1-h incubation period, to determine their stability under acidic and alkaline conditions. PA4 and PA8 exhibited good stability between pH 3–11, with a minimal loss in viability (0.1–1.96 logs PFU mL^−1^) after 1 h, and no significant decrease compared to the viability at pH 7 detected (Appendix A). Compared to the values at pH 7, phages displayed a significant decrease in stability at pH 12 with no viable phage detected (*p* < 0.0001) (Appendix A). All of the other phages except PA1 and PA10 showed a similar pH stability compared with PA4 and PA8 (Appendix A). Among all of the phages, PA1 displayed a significant increase in viability (*p* < 0.04) at pH 10 (Appendix A) and PA10 displayed a significant decrease in viability at pH 8 (*p* < 0.0001) (Appendix A). PA11 was removed from further experiments following pH and temperature stability testing due to the formation of two phenotypically different plaque types and is therefore not included in the results.

### 3.4. Bacteriophage Thermal Stability 

PA4 was stable at temperatures between 4–70 °C and did not lose viability after 1 h of incubation at the respective temperatures (Appendix A). However, no plaques were detected at 80 °C, which suggests there is no active phage at that temperature (Appendix A). PA8 was stable at temperatures between 4–50 °C; however, the viability significantly reduced (*p* < 0.0001) at 60 °C and no phage plaques were observed at 70 °C and 80 °C (Appendix A).

Similarly, all of the remaining phages were stable between 4–50 °C, but no detectable plaques were observed at 80 °C (Appendix A). Phages PA1, 2, 3, 5, 6 and 7 remained stable at 60 and 70 °C, phages PA9, 12 and 13 showed a reduction in viability between 3.6 and 4.3 logs PFU mL^−1^ at 60 °C, whereas phages PA10, 14 and 15 showed no viability between 60–80 °C (Appendix A).

### 3.5. Long-Term Storage Stability of Bacteriophage

The phages were tested for stability in long-term storage over 6 months at RT, 4 °C, −20 °C and −80 °C (all in TSB). PA4 maintained viability in all of the conditions over 6months, with a minimal reduction in the phage titer (<1 log PFU mL^−1^) (Appendix A). PA8 displayed the greatest viability over 6 months when stored at −80 °C (Appendix A). The viability was also maintained at 4 °C with < 1 log PFU mL^−1^ loss over 6 months, and a loss of 1.3 log PFU mL^−1^ at −20 °C (Appendix A). PA8 maintained good stability for the first 3 months at RT; however, a reduction in 2.2 log PFU mL^−1^ was observed after 6 months of storage (Appendix A).

Overall, the viability decreased the most for all of the phages when stored at RT compared to the other temperatures, and storage at −80 °C demonstrated the most stable condition for phage viability over a long time period (Appendix A). Phage 13, 14 and 15 demonstrated a reduction in the phage titers over 5 months, with no viable phage detected for PA14 and PA15 when stored at RT for 5 months and for all three phages when stored at RT for 6 months (Appendix A). 

### 3.6. Bacteriophage Adsorption Assays

The phage planktonic binding properties were determined through adsorption assays. Two phages, PA4 and PA8, which displayed broad but varied sensitivity to the 20 *P. aeruginosa* clinical isolates, were selected to maximise the target host range. A selection of 1 CF (C462) and 1 non-CF (C446) CI will be analysed in the results along with the PA01 strain.

PA4 showed a strong adsorption to isolates with a minimum of 4% of unadsorbed phage at 40 min, compared to PA8 with a minimum of 16% of unadsorbed phage (Figure 2b,f). Fifty percent of adsorbed phage in the total inoculum was achieved around 20 min for both PA4 and PA8 (Figure 2). 

### 3.7. Bacteriophage Inhibition Assays

The phage planktonic infective properties were determined through inhibition assays. The phage inhibition of reference strain PA01 and clinical isolates C446 and C462 was assessed over 6.5 h when infected at an MOI = 0.1 and 1 compared to the untreated control. PA4 showed significant inhibition of PA01 at an MOI = 0.1 (*p* < 0.0002) and an MOI = 1 (*p* < 0.0001) (Figure 3a). Similarly, a significant reduction of C446 growth was observed at an MOI = 0.1 (*p* < 0.02) and an MOI = 1 (*p* < 0.0002) (Figure 3b). PA4 did not significantly reduce the growth of C462 in planktonic form over 6.5 h (Figure 3c). PA8 did not significantly inhibit the growth of the three isolates over 6.5 h, demonstrating a poor infective ability (Figure 3d–f).

### 3.8. Crystal Violet Biofilm Assay

The three isolates (1 CF: C462, 1 non-CF: C446 and 1 PA01 reference strain) analysed in the planktonic adsorption assays were assessed using the microtiter dish biofilm assay. The absorbance readings for the crystal violet-stained biofilm assays after 24 h of treatment are displayed in Figure 4. PA4 significantly reduced the biomass of C446 at 1 × 10^7^ (*p* < 0.008), 1 × 10^8^ (*p* < 0.03) and 1 × 10^9^ (*p* < 0.01) PFU mL^−1^ each time compared to untreated control (Figure 4b). A significant reduction of C462 by PA4 was also observed at 1 × 10^7^ (*p* < 0.04), 1 × 10^8^ (*p* < 0.01) and 1 × 10^9^ (*p* < 0.009) PFU mL^−1^ (Figure 4c). PA4 and did not reduce the biomass of PA01 compared to the untreated control (Figure 4a,d). PA8 showed no significant reduction in biomass for all three of the isolates when treated for 24 h (Figure 4d–f). 

### 3.9. LIVE/DEAD Assay

LIVE/DEAD *Bac*Light Bacterial Viability Kit was used to determine the percentage of dead (PI, red) and live (SYTO9, green) bacteria in the samples representative of each treatment group for PA4-treated biofilms compared to the control (Appendix A). A statistically significant increased percentage of dead/live cells in PA4-treated biofilms compared to the untreated control were observed in all three assays (Figure 5). 

### 3.10. Bacteriophage Morphology 

The electron micrographs showed that both of the phages had an icosahedral head with a non-contractile tail (Figure 6). PA4 (A) is characteristic of the Myoviridae family, whilst PA8 (B) is characteristic of the Podoviridae family (Figure 6).

### 3.11. Genomic Analysis of Bacteriophage

The complete genome sequence of *P. aeruginosa* PA4 has been deposited in the NCBI GenBank database and assigned accession number MZ285878 along with the PA8 accession number MZ285879. The PA4 sequence length was 66,450 bp with a GC (guanine-cytosine) content of 55.6% and the PA8 sequence length was 63,654 bp with a GC content of 63.55%. A basic local alignment search tool (BLAST) analysis of PA4 indicates it belongs to the Myoviridae family, whilst PA8 belongs to the Podoviridae family. PA4 was predicted to possess 97 open read frames (ORFs), of which 19 ORFs have known protein functions. Two ORFs encoded cell lysis-associated proteins. No lysogenic genes or virulence-associated, toxic or antibiotic resistance genes were identified. PA4 has a ~97–98% similarity to other phages in the database (Figure 7a). PA8 showed ~98% similarity to the genome of a variety of *P. aeruginosa* strains across its range of coverage (73–93% depending on the strain) and a ~98% similarity (across a coverage range of 84%) to a group of *Pseudomonas* phages that were specifically chosen for their temperate nature [29] (Figure 7b). Additionally, a genomic annotation revealed the presence of the recombinase gene *rdgC* and an integrase gene in PA8, often present in temperate phages [8].

## 4. Discussion

A *P. aeruginosa* airway infection is associated with worsening lung function in CF patients [2]. Treatment with systemic and inhaled broad-spectrum antibiotics from early in the life of those patients can lead to the development of antibiotic-resistant strains of *P. aeruginosa* [2,3,4]. Due to the emerging threat of MDR infections and lack of research and development of new antibiotics, there is an urgent need for new therapeutic treatment options [6]. *P. aeruginosa* phages have shown to be effective against planktonic and biofilm *P. aeruginosa*; however, clinical-grade phages are currently not available in Australia [1,4,14,30].

In this study, *P. aeruginosa* phages isolated from wastewater displayed a broad host range to the 20 clinical isolates tested (9 CF and 11 non-CF), with a total of 18/20 (90%) isolates susceptible (sensitive or semi-sensitive) to at least 1/15 phages. The infective properties of two selected phages (PA4 and PA8) demonstrated variability in bacterial killing, and therefore the potential to increase the target host range when combined in a cocktail and were selected for further analysis. One particular concern with the use of phages is indeed the emergence of phage-resistant bacteria [18,31]. The combination of phages in cocktails that are regularly changed with new or different phages could potentially address this issue by maintaining selective pressure on the bacterial host [31].

Phage applications in humans require them to be stable in acidic, alkaline and high- or low-temperature conditions. In a therapeutic setting, pH levels for the administration of phages need to be considered [4]. For example, oral administration of phages in a tablet form would pass through a very acidic environment in the gastrointestinal tract [4]. A nasal spray would expose the phages to a different environment with pH levels around 5.5–6.5 dependent on the nasal mucosa [4]. Results from this study show phages were stable at a wide pH range (3–11). 

Temperature is an important factor in phage survivability and plays a significant role in phage infective properties, such as attachment and replication [32]. Phage activity at different temperatures varies amongst strains, and studies have shown that high temperatures can lead to nucleic acid and protein denaturation and an inactivate phage [31]. In the current study, the phages were viable between 4–50 °C with seven of the fifteen phages maintaining viability at 60 °C and 70 °C; however, exposure to 80 °C for 1 h inactivated all of the phages. This variation may be due to different strains being more sensitive to higher temperatures, or phages developing heat resistance through mutations [31]. 

The long-term storage and stability of selected phages is desirable, and a good candidate for phage therapy should be one that maintains an infective ability upon storage [31]. The phage long-term storage viability was determined at RT, 4 °C, −20 °C and −80 °C to ensure their integrity during storage and transfer [31]. The phages survived well at −80 °C, with minimal activity loss over 6 months of storage. The phage activity at 4 °C and −20 °C demonstrated similar results with a greater reduction in the viability compared to −80 °C; however, the viability of seven phages at RT began to decline after 1month of storage. This is in line with previous studies which suggested the storage of phages at RT is recommended for no longer than 40 days and long-term storage of phages is recommended at −80 °C [32].

The efficacy of phages to infect planktonic bacterial cells was measured using phage adsorption and inhibition assays. The phage binding properties varied amongst clinical isolates; however, PA4 demonstrated a strong attachment to the target pathogen, suggesting it had efficient binding properties towards the host surface. This phage, belonging to the Myoviridae family, varies substantially in its structure compared to PA8, belonging to the Podoviridae family, as confirmed by TEM. The activity of PA4 showed an inhibitory effect towards clinical isolates over 6.5 h, unlike PA8, which showed negligible effects. Sequencing of these phages revealed the lytic ability of PA4, which supports these findings, whilst PA8 is characteristic of a temperate phage and instead has the capacity to integrate into the bacterial host genome. This phage was therefore excluded for inclusion in future clinical-grade phage cocktails.

One of the main advantages of phage treatments is their activity against MDR pathogens and their supposed efficacy against biofilms [2]. In this study, some of the clinical isolates (e.g., C458) were resistant to at least two classes of commonly used antibiotics in the treatment of *P. aeruginosa* infection and 14/20 of the isolates (including C458) were sensitive or semi-sensitive to PA4. The lytic effect of PA4 on planktonic bacteria translated well to a reduction in the biofilm biomass, given the strictly lytic nature of the phage. This is consistent with previous studies showing the efficacy of phages to reduce biofilm in vitro and *in vivo* [17]. The results show a slightly better killing of the C462 strain at 1 × 10^9^ PFU mL^−1^ after 24 h of treatment. However, the ability of PA4 to reduce the biomass of the C446 strain tended to be greater at a lower concentration. This is supported by other studies of *P. aeruginosa* phages, which have shown that a higher concentration does not necessarily translate to a greater antibiofilm activity [33,34]. This non-linear relationship may be expected for some phages due to their self-replicating nature [33]. The advantage of screening phages at various concentrations is the possibility of identifying the lowest acceptable concentration range since the application of phages at large volumes in a therapeutic setting may be problematic and costly [33]. Therefore, the ability of PA4 to effectively reduce the biomass at a low concentration makes it a suitable candidate for phage therapy. This is demonstrated in whole by its broad infectivity profile, good stability in various pH and temperature conditions, lytic ability and the absence of antibiotic-resistant, toxic or lysogenic genes.

## 5. Conclusions

This study provides a small library of phages targeting CF and non-CF *P. aeruginosa*. One strictly lytic phage was identified as a potential therapeutic candidate with good stability and a broad host range. The isolation and characterisation of further phages is needed in order to be incorporated into phage cocktails targeting *P. aeruginosa* prior to the translation towards the clinical application. 

## Figures and Tables

**Figure 1 microorganisms-09-02001-f001:**
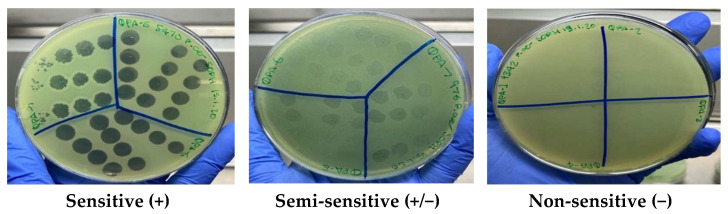
Bacteriophage Host Range Spot Assay. Double agar spot assay demonstrating the three clarities of plaques for phage sensitivity: sensitive (+), semi-sensitive (+/−) or non-sensitive (−) on the lawn of *P. aeruginosa* clinical isolates C420, C413 and C422 respectively.

**Figure 2 microorganisms-09-02001-f002:**
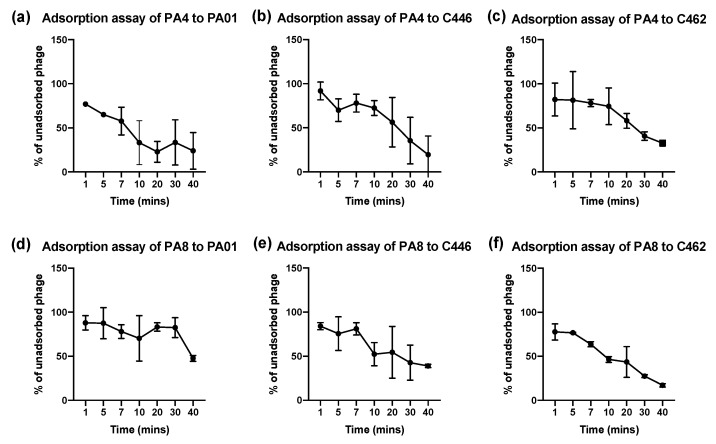
Determining Percentage of Unadsorbed Bacteriophage to Host Bacterial Surface. Plaque forming units per mL (PFU mL^−1^) was measured for PA4 and PA8 at 1 min, 5 min, 7 min, 10 min and every 10 min thereafter for 40 min against PA01 reference strain (**a**,**d**), 1 non-CF (C446) CI (**b**,**e**) and 1 CF (C462) CI (**c**,**f**) to determine the percentage of unadsorbed phage of the total phage inoculum. Data expressed as mean ± SEM for three independent experiments.

**Figure 3 microorganisms-09-02001-f003:**
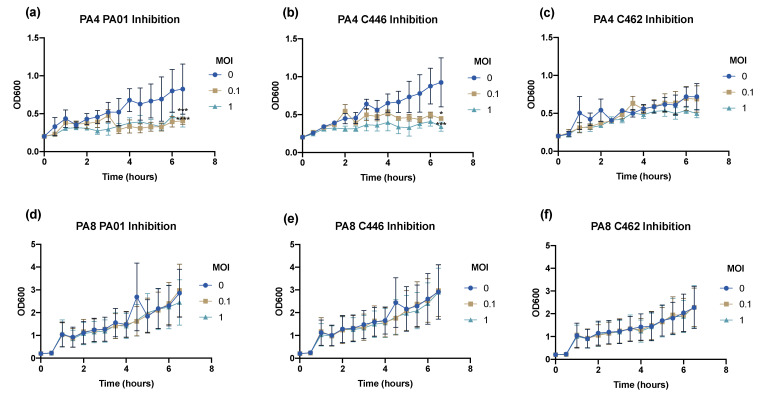
Bacteriophage Inhibition Assays. Inhibition of 1 CF (C462), 1 non-CF (C446) clinical isolate and PA01 reference strain after treatment with PA4 (**a**–**c**) and PA8 (**d**–**f**) at a multiplicity of infection (MOI) = 0.1 and 1 compared to untreated control. Data expressed as mean ± SEM for three independent experiments. *, *p* < 0.05; ***, *p* < 0.001; ****, *p* < 0.0001.

**Figure 4 microorganisms-09-02001-f004:**
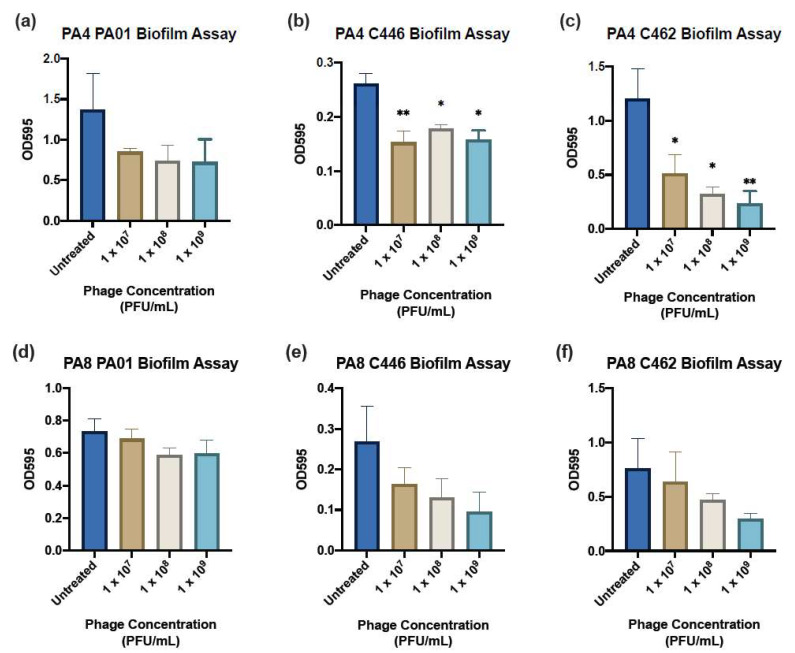
Biofilm Assays. Biomass of biofilms as measured by absorbance at 595 nm of crystal violet-stained biofilms after 24 h of treatment with PA4 (**a**–**c**) and PA8 (**d**–**f**) at three different concentrations (1 × 10^7^, 1 × 10^8^, 1 × 10^9^ PFU mL^−1^) with significance determined compared to the untreated control. Data expressed as mean ± SEM for three independent experiments. *, *p* < 0.05; **, *p* < 0.01.

**Figure 5 microorganisms-09-02001-f005:**
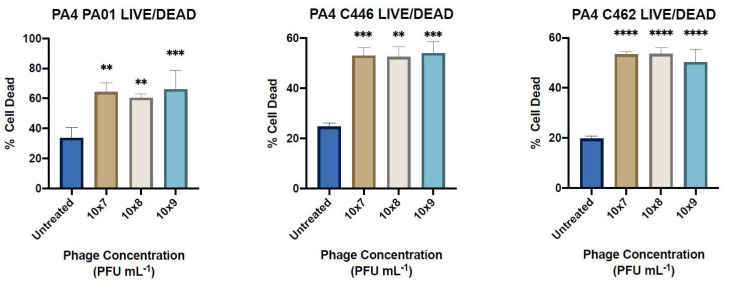
LIVE/DEAD Assays. Percentage of cells dead in biofilms treated with PA4 at three different concentrations (1 × 10^7^, 1 × 10^8^, 1 × 10^9^ PFU mL^−1^) with significance determined compared to the untreated control. Data expressed as mean ± SEM for three independent experiments. **, *p* < 0.01; ***, *p* < 0.001; ****, *p* < 0.0001.

**Figure 6 microorganisms-09-02001-f006:**
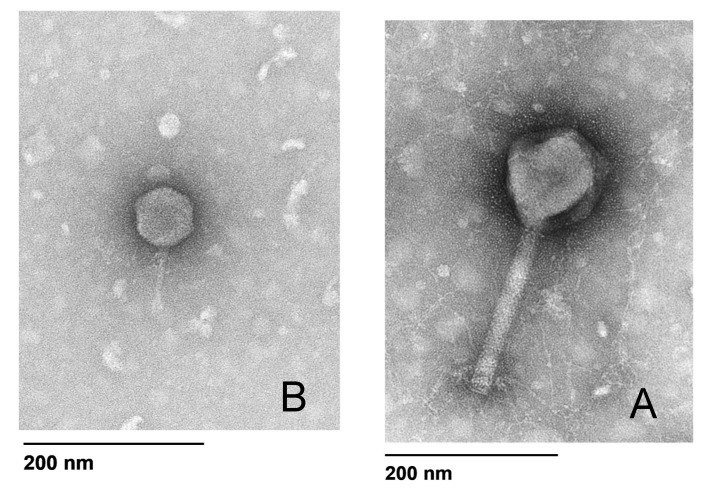
Transmission electron micrographs. Transmission electron micrograph of (**A**) PA4 belonging to Myoviridae family and (**B**) PA8 belonging to Podoviridae family.

**Figure 7 microorganisms-09-02001-f007:**
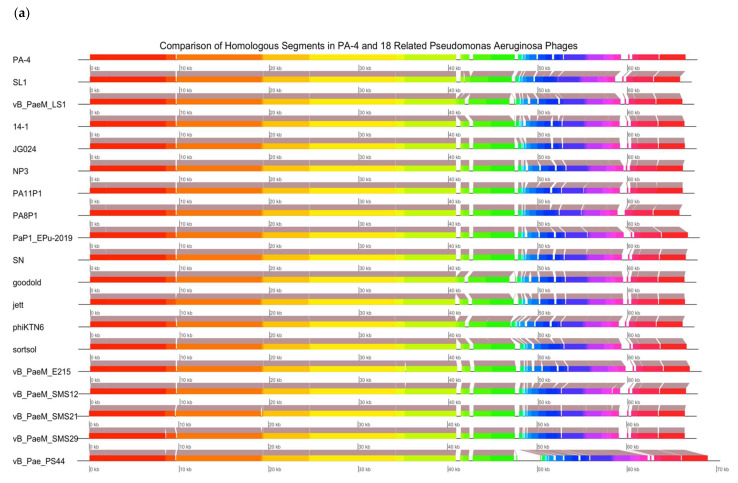
Bacteriophage Genome Sequence Analysis. Schematic genome alignment diagram of (**a**) PA4 and 18 closely related *P. aeruginosa* phages and (**b**) PA8 and 14 closely related *P. aeruginosa* phages, obtained using the Mauve software package and plotted using genoPlotR. Phages in figure (**a**) were oriented to begin at the *thyx* gene, whilst phages in figure (**b**) were oriented to begin at the *rdgC* recombinase gene to assist the visual interpretation. The coloured blocks are homologous nucleotide regions, whereas the white spaces and gaps indicate regions that are different between phages.

**Table 1 microorganisms-09-02001-t001:** Summary of Bacteriophage Host Range. * Based on the degree of lysis on the bacterial lawn, the spots were differentiated into three categories: sensitive (+), semi-sensitive (+/−) or non-sensitive (−). C450 to C457 represent *P. aeruginosa* strains isolated from cystic fibrosis patients whilst C420 to C432 represent *P. aeruginosa* strains isolated from non-cystic fibrosis patients.

Host Range * of Phages against *P. aeruginosa* Clinical Isolates														
*P. aeruginosa* *Strain*	PA1	PA2	PA3	PA4	PA5	PA6	PA7	PA8	PA9	PA10	PA12	PA13	PA14	PA15
**C450**	**+**	**+**	**+**	**+**	**+**	**+**	**+**	**+/−**	**+/−**	**+/−**	**+/−**	**+/−**	**+/−**	**+/−**
**C460**	**+**	**+**	**+**	**+**	**+**	**+**	**+**	**+**	**+**	**+**	**+**	**+**	**+**	**+**
**C462**	**+/−**	**+/−**	**+/−**	**+/−**	**+/−**	**+/−**	**+/−**	**−**	**−**	**−**	**−**	**−**	**−**	**−**
**C453**	**−**	**−**	**−**	**−**	**−**	**−**	**−**	**−**	**−**	**−**	**−**	**−**	**−**	**−**
**C458**	**+**	**+**	**+**	**+**	**+**	**+**	**+**	**+/−**	**+/−**	**+/−**	**+/−**	**+/−**	**+/−**	**+/−**
**C459**	**+**	**+**	**+**	**+**	**+**	**+**	**+**	**+**	**+**	**+**	**+**	**+**	**+**	**+**
**C462**	**+**	**+**	**+**	**+**	**+**	**+**	**+**	**+**	**+**	**+**	**+**	**+**	**+**	**+**
**C452**	**−**	**−**	**−**	**−**	**−**	**−**	**−**	**−**	**−**	**−**	**−**	**−**	**−**	**−**
**C457**	**−**	**+**	**+**	**−**	**+**	**+/−**	**+**	**+**	**+**	**+**	**+**	**+**	**+**	**+**
**C420**	**+**	**+**	**+**	**+**	**+**	**+**	**+**	**+/−**	**+/−**	**+/−**	**+/−**	**+/−**	**+/−**	**+/−**
**C446**	**+**	**+**	**+**	**+**	**+**	**+**	**+**	**+/−**	**+/−**	**+/−**	**+/−**	**+/−**	**+/−**	**+/−**
**C410**	**+/−**	**+/−**	**+/−**	**+/−**	**+/−**	**+/−**	**+/−**	**+/−**	**+/−**	**+/−**	**+/−**	**+/−**	**+/−**	**+/−**
**C513**	**+/−**	**+/−**	**+/−**	**+/−**	**+/−**	**+/−**	**+/−**	**+/−**	**+/−**	**+/−**	**+/−**	**+/−**	**+/−**	**+/−**
**C438**	**+**	**+**	**+**	**+**	**+**	**+**	**+**	**+/−**	**+/−**	**+/−**	**+/−**	**+/−**	**+/−**	**+/−**
**C396**	**−**	**−**	**−**	**−**	**−**	**−**	**−**	**+**	**+**	**+**	**+**	**+**	**+**	**+**
**C422**	**−**	**−**	**−**	**−**	**−**	**−**	**−**	**+/−**	**+/−**	**+/−**	**+/−**	**+/−**	**+/−**	**+/−**
**C434**	**+**	**+**	**+**	**+**	**+**	**+**	**+**	**+/−**	**+/−**	**+/−**	**+/−**	**+/−**	**+/−**	**+/−**
**C440**	**−**	**+/−**	**+/−**	**−**	**+**	**−**	**+**	**+/−**	**+/−**	**+/−**	**+/−**	**+/−**	**+/−**	**+/−**
**C423**	**+**	**+**	**+**	**+**	**+**	**+**	**+**	**+**	**+**	**+**	**+**	**+**	**+**	**+**
**C432**	**+/−**	**+/−**	**+/−**	**+/−**	**+/−**	**+/−**	**+/−**	**+/−**	**+/−**	**+/−**	**+/−**	**+/−**	**+/−**	**+/−**

## Data Availability

Not applicable.

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
