# Peer review of "Preclinical Development of a Bacteriophage Cocktail for Treating Multidrug Resistant Pseudomonas aeruginosa Infections"

_microorganisms, 2021, doi:10.3390/microorganisms9092001_

Round 1

Reviewer 1 Report

  • My compliment for the extensive description of the work-up which has to be done for the characterization of the phage isolates and their effectivity. This is a basic necessity before any of them can be applied clinically. In addition, the development of clinical phage therapy is in great need of such sound methodological analyses and methods in order to convince the clinical world of the potency by good and safe clinical practices.
  • In this respect, I strongly advice to refer somewhere to the paper by Pirnay et al ( 2015) on Quality and Safety Requirements for sustainable phage therapy ( Pharm Res (2015) 32:2173-2179. I was a little surprised that this paper is not mentioned, it gives the consent by a large group of phage and phage therapy experts on the minimal requirements before clinical application of phages. Of course, the present article solves this in the title by limiting the width of their work to  "Preclinical development...". Nonetheless the authors should mention this consensus paper and maybe indicate how their own set of tests already fulfills the consented requirements for clinical use, or which further tests would be necessary before application of their cocktail or separate phages in it.
  • The PA4 phage is a single favorable candidate which has good to excellent perspectives for effective clinical application. With only one such top-phage it is a bit meager to state in the Conclusion on line 480 that the study provides a library of therapeutic phages.... Yes, the authors' extensive characterization by the many tests makes all phages which they tested suited to be listed in a library. But when you name it a library of candidate therapeutic phages, I think it would be better to name it at least "a small library" or "interesting collection". I underline that this is just a minor comment about only one single wording, but I think that is what I and other readers might also feel...It does not all criticize the authors' good work, which I see as almost part of a laboratory handbook for characterization of wild phages isolated from a hospital's wastewater or any other source.
  • In the acknowledgement I missed the Otorhinolaryngology centre in Amsterdam, which donated the P. strains. Though this donator is already mentioned under Materials and Methods ( "kindly donated"), but it is appropriate to (re)mention it also in the specific Acknowledgement paragraph. Btw, I have totally none connection with the respective centre.

Reviewer 2 Report

The manuscript concerns a Bacteriophage cocktail active against antibiotic 
resistant P. aeruginosa strains. 

In my opinion, only one thing should be added; The LIVE/DEAD microscope immage of the experiment, in order to stress the result.

Round 2

Reviewer 2 Report

The work can be published in this form